# Spermidine as a Potential Protective Agents Against Poly(I:C)-Induced Immune Response, Oxidative Stress, Apoptosis, and Testosterone Decrease in Yak Leydig Cells

**DOI:** 10.3390/ijms26062753

**Published:** 2025-03-19

**Authors:** Yujun Tang, Hao Li, Yutian Zeng, Cuiting Yang, Run Zhang, Arab Khan Lund, Ming Zhang

**Affiliations:** 1College of Animal Science and Technology, Sichuan Agricultural University, Chengdu 611130, China; 13171290208@163.com (Y.T.); 17844611985@163.com (H.L.); alisaz1996@163.com (Y.Z.); yangcuiting922@outlook.com (C.Y.); zhangrun202111@163.com (R.Z.); aarabbaloch@gmail.com (A.K.L.); 2Faculty of Animal Production and Technology, Shaheed Benazir Bhutto University of Veterinary and Animal Science, Sakrand 67210, Pakistan; 3Key Laboratory of Livestock and Poultry Multi-Omics, Ministry of Agriculture and Rural Affairs, College of Animal Science and Technology, Sichuan Agricultural University, Chengdu 611130, China; 4Farm Animal Genetic Resources Exploration and Innovation Key Laboratory of Sichuan Province, College of Animal Science and Technology, Sichuan Agricultural University, Chengdu 611130, China

**Keywords:** spermidine, Leydig cell, immune response, oxidative stress, yak

## Abstract

Viral infections of the reproductive tract and testis in male yaks, often resulting from natural mating under grazing conditions, can lead to infertility due to Leydig cell (LC) apoptosis, immune activation, oxidative stress, and reduced testosterone production. Spermidine (SPD), a potential therapeutic agent with antioxidant and anti-aging properties, might alleviate oxidant stress, immune response, and virus infection caused by apoptosis. In this study, firstly testicular Leydig cells of yak were induced with Poly(I:C), the pathogen-associated molecular pattern of the dsRNA virus, as a pathogenic model at the cellular level. Secondly, immune response, apoptosis, oxidative stress, and testosterone synthesis were measured in LC with or without SPD culture medium. Finally, transcriptomic sequencing was utilized to investigate the molecular mechanisms underlying the protective effects of SPD. These results suggested Poly(I:C) damaged the function of Leydig cells, significantly decreased the concentration of testosterone, and induced immune response, oxidative stress, and cell apoptosis, while SPD significantly alleviated the immune response and oxidative stress, and then significantly inhibited cell apoptosis and restores testosterone production in LCs. Transcriptomic analysis revealed that SPD significantly alleviates inflammation and apoptosis induced by Poly(I:C), reducing immune response and cellular damage through the regulation of several key gene expressions. These findings suggest SPD has the potential ability to mitigate Poly(I:C)-induced immune response, oxidative stress, and apoptosis, and then restore testosterone production in Leydig cells, offering a promising strategy to protect and enhance male yak fertility after infection with dsRNA virus.

## 1. Introduction

Yaks (*Bos grunniens*) are essential livestock in the Qinghai–Tibet Plateau and other high-altitude regions, where they have adapted to harsh environmental conditions. However, they face significant reproductive challenges, including low fertility rates, which are exacerbated by viral infections and uncontrolled mating under grazing conditions. These reproductive issues pose a serious threat to yak farming, which is vital for the livelihoods of local populations. Leydig cells (LCs), located in the interstitial tissue of the testes, play a crucial role in regulating male reproductive health by producing testosterone and regulating spermatogenesis [1]. Beyond their primary function in steroidogenesis, LCs are involved in the immune response, secreting cytokines and other factors that help maintain cellular homeostasis and protect against infections [2]. However, the function of Leydig cells can be compromised by oxidative stress, inflammation, and viral infections, leading to impaired fertility.

In addition to supporting spermatogenesis [3], testosterone plays a vital role in maintaining the blood–testis barrier, regulating libido, and supporting overall male fertility [2]. Testosterone secretion in Leydig cells is regulated by luteinizing hormone (LH), which stimulates a signaling cascade through cyclic adenosine monophosphate (cAMP), triggering cholesterol transport and its conversion to pregnenolone, the precursor of testosterone [4,5,6]. During fetal development, testosterone is also crucial for the morphogenesis of the genital tract and the descent of the testes [7]. However, when Leydig cells are exposed to environmental stressors, such as oxidative damage or viral infection, their ability to produce testosterone and support spermatogenesis is impaired. This is particularly problematic in high-altitude regions, where hypoxia and immune challenges can exacerbate reproductive dysfunction.

The male yak (*Bos grunniens*), as an economically important animal in plateau regions, faces significant threats to its reproductive health from viral infections. With climate change, environmental pollution, and the expansion of farming scale, viral diseases have become one of the major factors limiting yak productivity. Viral infections not only suppress immune system function but may also cause direct damage to the male reproductive system, thereby impairing fertility. In studying the immune responses and reproductive functions of male reproductive organs, the cellular responses induced by the double-stranded RNA (dsRNA) viral mimic Poly(I:C) represent a critical research direction. Poly(I:C), a synthetic dsRNA analog, is widely used in research models to simulate viral infections and study immune responses. The rationale for choosing Poly(I:C) lies in its ability to effectively activate pattern recognition receptors (e.g., TLR3 [8] and MDA5), triggering immune responses similar to those induced by natural dsRNA viruses, including the release of inflammatory cytokines (e.g., IL-6 and TNF-α) and the activation of apoptotic pathways [9]. Furthermore, Poly(I:C) exhibits high stability and reproducibility, making it a reliable experimental platform for investigating the mechanisms of viral infection-induced damage to the male reproductive system. Poly(I:C) mimics viral infection by activating Toll-like receptor 3 (TLR3) and increasing the expression of immune cytokines, including IL-6 and members of the poly ADP-ribose polymerase (PARP) family [10], leading to immune activation and apoptosis in many cell types, including epithelial cells of the genital tract [5]. However, little is known about its impact on Leydig cells, which are crucial for testosterone production and overall male fertility. Given that viral infections and inflammation are key contributors to impaired reproductive health in yaks, understanding the effects of Poly(I:C) on Leydig cells could provide valuable insights into the mechanisms underlying male infertility in this species and other livestock.

Spermidine (SPD), a naturally occurring polyamine, has garnered attention for its potent antioxidant, anti-inflammatory, and autophagy properties [9,11]. SPD has been shown to induce autophagy, reduce oxidative stress, and protect cells from apoptosis in various disease models, including neurological disorders, cardiovascular diseases, and cancer [12,13]. More recently, SPD has been investigated for its potential to mitigate cellular damage caused by inflammation and oxidative stress, both of which are major contributors to male infertility. In humans, white geese, mice and yeast nematodes (*Caenorhabditis elegans*), and flies (*Drosophila melanogaster*), SPD has been shown to play a critical role in maintaining cellular homeostasis, inducing autophagy and mitigating stress-induced damage [14]. For instance, SPD induces autophagy, a process essential for cellular clearance and survival, and scavenges reactive oxygen species (ROS) to protect cells from oxidative stress [15]. Additionally, SPD suppresses pro-inflammatory cytokines (e.g., IL-6, TNF-α), highlighting its role in modulating immune responses [16]. However, species-specific differences exist in SPD metabolism and physiological effects. Variations in the expression and activity of key enzymes (e.g., spermidine synthase, spermine synthase) and tissue-specific responses, particularly in reproductive tissues, may influence the efficacy and functional outcomes of SPD [17]. These similarities and differences underscore the universality of SPD’s biological roles while emphasizing the need for species-specific investigations to fully elucidate its mechanisms and therapeutic potential. Although some preliminary studies have been conducted on spermidine in the context of male reproduction, research in this area remains in its early stages, and many aspects of its role are still not fully understood. Its potential to protect Leydig cells from the dual stress of oxidative damage and immune activation, such as that induced by Poly(I:C), remains unexplored.

This study aims to fill this gap by investigating the therapeutic potential of spermidine in protecting Leydig cells from oxidative stress, apoptosis, and functional damage induced by Poly(I:C). We hypothesize that SPD may exert its protective effects by modulating oxidative stress pathways, enhancing autophagy, and regulating immune responses in Leydig cells. Through this, SPD could potentially restore testosterone production and support spermatogenesis, offering a novel approach to mitigate the impact of viral-induced stress on male fertility.

The significance of this study extends beyond the specific case of yaks. Given the increasing challenges posed by infectious diseases and environmental stressors to livestock fertility worldwide, our findings could have broader implications for improving male reproductive health in other species. By exploring the protective mechanisms of SPD in Leydig cells, this research could contribute to the development of novel strategies for enhancing fertility in livestock under challenging environmental conditions, providing valuable insights for animal breeding programs and agricultural practices.

## 2. Results

### 2.1. In Vitro Toxicity Validation of SPD

To assess the safety profile of spermidine (SPD), we conducted an in vivo toxicity evaluation in mice (Figure 1A). After a seven-day acclimatization period, the mice were randomly divided into two groups: one group received regular drinking water, while the other group was provided with drinking water supplemented with SPD at a dose of 10 mg/kg. The experiment was carried out over 1, 7, and 14 days to evaluate potential toxic effects.

The hematological examination results are shown in Figure 1A–H. The levels of alanine aminotransferase (ALT) and aspartate aminotransferase (AST) in the SPD-treated group did not show significant increases on days 1, 7, and 14, indicating that SPD treatment did not cause obvious liver damage (Figure 1A,B). Notably, on days 7 and 14, the AST and ALT levels in the SPD-treated group were significantly lower than those in the control group (*p* < 0.05, *p* < 0.01), suggesting that SPD may have a protective effect without inducing toxicity.

For hematological parameters such as red blood cell count (RBC), hemoglobin (HGB), hematocrit (HCT), mean corpuscular volume (MCV), mean corpuscular hemoglobin (MCH), and mean corpuscular hemoglobin concentration (MCHC) (Figure 1C–H), there were no significant differences between the SPD-treated group and the control group at any time point (*p* > 0.05). This indicates that SPD did not adversely affect the hematopoietic system of the mice, and no signs of anemia or other hematological abnormalities were observed. Furthermore, no significant fluctuations were observed in MCV, MCH, or MCHC, further confirming that SPD did not negatively impact erythrocyte-related functions in the mice.

Figure 1I illustrates the histological changes in the major organs (heart, liver, spleen, lungs, and kidneys) of the mice. Tissue sections were examined using hematoxylin and eosin (H&E) staining. The results revealed no significant morphological changes or tissue damage in the major organs of the SPD-treated group. All tissues exhibited intact structures with regular cell arrangements, and no signs of cellular edema, necrosis, or inflammatory responses were observed. In particular, the liver and kidney tissue sections of the SPD-treated group showed no significant differences compared to the control group, indicating that SPD did not cause damage to the liver, kidneys, or other vital organs.

In conclusion, SPD at a dose of 10 mg/kg did not cause significant toxicity or damage to the physiological health or organ function of the mice. No adverse changes were observed in hematological or biochemical parameters, and H&E staining revealed no abnormalities in organ tissue structures. These findings demonstrate that SPD is safe for mice and does not significantly impact their physiological state or tissue health, supporting its safety for use.

### 2.2. Poly(I:C) Induces Immune Response, Apoptosis, and Functional Damage in LCs

To determine if Poly(I:C) induces an immune response, LCs were treated with a constant concentration of 0.5 μg/mL of Poly(I:C) for 6, 12, and 24 h, and 3, 6, and 12 h. Notably, IL-6 and tumor necrosis factor alpha (TNF-α) levels increased significantly after 6 h (Figure 2Aa,Ac; *p* < 0.05). Further, we exposed LCs to different concentrations (0.1, 0.5, and 1 μg/mL) for 6 h, revealing that IL-6 and TNF-α showed the greatest increase at 0.5 μg/mL (Figure 2Ab,Ad; *p* < 0.01). Based on the results from dose-response and time-course experiments, IL6 and TNFα gene expression levels were significantly upregulated when cells were treated with Poly(I:C) at 0.5 µg/mL for 6 h; therefore, we ultimately determined these conditions as the optimal concentration and treatment duration. Next, we treated LCs with 0.5 μg/mL Poly(I:C) for 6 h to assess the LCs apoptosis. The apoptosis rate of Leydig cells was significantly increased in the Poly(I:C) group compared to the control group (Figure 2B; *p* < 0.05). We then conducted an enzyme-linked immunosorbent assay (ELISA) to measure SOD, MDA, and testosterone levels. The Poly(I:C) treatment resulted in a notable increase in MDA and a decrease in SOD (Figure 2C; *p* < 0.05 and Figure 1D; *p* < 0.001), as well as a significant reduction in testosterone levels (Figure 2E; *p* < 0.05). The results suggest that Poly(I:C) stimulation can trigger an immune response, induce apoptosis, elevate oxidative stress in LCs, and impair testosterone secretion.

### 2.3. SPD Attenuates Poly(I:C)-Induced Immune Response in Leydig Cell

To investigate whether SPD can alleviate Poly(I:C)-induced immune response in LCs, we assessed the expression of immune response-related genes in the Poly(I:C) and SPD + Poly(I:C) group. The results indicate that SPD significantly reduced the upregulation of IL-6 (*p* < 0.05) (Figure 3A), interleukin-8 (IL-8) (*p* < 0.01)(Figure 3B), TNF-α (*p* < 0.05) (Figure 3D), P65 (*p* < 0.01) (Figure 3E), REL (*p* < 0.0001) (Figure 3F), and MAPK (*p* < 0.01) (Figure 3G)induced by Poly(I:C). To further investigate the role of SPD in mitigating Poly(I:C)-induced immune response in LCs, Western blot analysis was conducted to assess the levels of immune response-related proteins IL-6, p65, and phosphorylation p65. The results indicated that the upregulated expression of these proteins was significant in the Poly(I:C) group and was notably reduced upon the addition of SPD (Figure 3H; *p* < 0.05). These findings indicate that SPD can reverse the immune response in Leydig cells induced by Poly(I:C).

### 2.4. SPD Recuses Poly(I:C)-Induced LC Apoptosis

The result in Section 2.2 has demonstrated that Poly(I:C) induces apoptosis in LCs. To evaluate the protective role of SPD against this apoptosis, cell viability and the expression of apoptosis-related genes and proteins were measured (Figure 4). The results indicated that SPD treatment significantly enhanced LC viability compared to the Poly(I:C) group (Figure 4A; *p*< 0.05). Early apoptotic cells were labeled with Annexin V-FITC (Figure 4B), and then the apoptosis was measured using flow cytometry. The apoptosis rate in the control group was approximately 21%, while Poly(I:C) treatment significantly increased the apoptosis rate to 34% (Figure 4C; *p* < 0.05), indicating that Poly(I:C) induces apoptosis in testicular interstitial cells. However, when spermidine (SPD) was added in combination with Poly(I:C), the apoptosis rate decreased to 22%, which was significantly lower than that in the Poly(I:C) group (Figure 4C; *p* < 0.01). The result indicated SPD extremely significantly mitigated Poly(I:C)-induced apoptosis of LCs. Furthermore, SPD decreased the expression of the BAX (Figure 4D; *p* < 0.0001) gene while promoting BCL-2 (Figure 4E; *p* < 0.05) gene expression. SPD inhibited the upregulation of Caspase-6 (Figure 4F; *p* < 0.05) and Caspase-3 (Figure 4G; *p* < 0.001) caused by Poly(I:C), and also inhibited the upregulation of Caspase-3 protein induced by Poly(I:C). Above all, SPD alleviates Poly(I:C)-induced apoptosis via inhibiting the expression of BAX, Caspase-3, and Caspase-6, and upregulating the expression of BCL-2.

### 2.5. SPD Alleviated Poly(I:C)-Induced Oxidative Stress of LCs

To investigate whether SPD can mitigate the impact of Poly(I:C) on the oxidative stress of LCs, the concentration of oxidative stress-related enzymes, including Glutathione (GSH), Catalase (CAT), Superoxide dismutase (SOD), and the oxidation product Malondialdehyde (MDA) were measured using ELISA assay. Poly(I:C) increased the concentration of MDA and inhibited CAT, SOD, and GSH activities. Conversely, SPD significantly reduced the MDA increase (*p* < 0.01) and enhanced GSH (*p* < 0.001), SOD, and CAT (*p* < 0.0001) activities (Figure 5A–D). These results indicated that SPD has an antioxidative function and alleviates the Poly(I:C)-induced oxidative stress of LCs.

### 2.6. SPD Alleviated the Decrease in Testosterone Synthesis Caused by Poly(I:C) in LCs

The synthesis of testosterone is a complex biochemical process involving many enzymes, and the expression of Cytochrome P450 family 11 subfamily A member 1 (CYP11A1), 17 subfamily A member 1 (CYP17A1), and Steroidogenic acute regulatory protein (StAR) related to steroid synthesis and metabolism was measured using RT-qPCR (Figure 6). The results showed that the expression of CYP11A1 (Figure 6A; *p* < 0.05) and CYP17A1 (Figure 6B; *p* < 0.01), the testosterone synthesis-related enzyme, was significantly elevated in Poly(I:C) group compared to the control group while the expression of StAR (Figure 6C; *p* < 0.01) and the concentration of testosterone (Figure 5D, *p* < 0.0001) was significantly decreased in Poly(I:C) group compared to the control group (*p* < 0.005). The comparison between SPD + Poly(I:C) and Poly(I:C) group indicated SPD alleviated the downregulation of StAR induced by Poly(I:C). SPD significantly alleviated the decrease in the concentration of testosterone induced by Poly(I:C) (Figure 6D; *p* < 0.01). These findings suggest that SPD might rescue the cellular function of Leydig cells secreting testosterone.

### 2.7. Expression Trends and Pathway Enrichment Analysis

To investigate the molecular mechanisms by which SPD alleviates Poly(I:C)-induced immune response, oxidative stress, and apoptosis while restoring testosterone production, transcriptome sequencing was performed on yak Leydig cells from the control, Poly(I:C), and Poly(I:C) + SPD groups. Differentially expressed genes (DEGs) from the three groups were analyzed to generate a Venn diagram, identifying 768 co-expressed genes (Appendix A), followed by trend analysis of these co-expressed DEGs (Figure 6A). Trends analysis revealed that 84 genes in clusters 1, 2, 3, and 6 were significantly downregulated after Poly(I:C) treatment, while SPD treatment restored their expression (Figure 7A), and KEGG analysis showed that these genes were enriched in these signaling pathway related to calcium, phospholipase D, Resin, and Wnt (Figure 7B). In contrast, 627 genes in clusters 4, 5, 9, and 10 were markedly upregulated by Poly(I:C) and suppressed by SPD (Figure 7A), and these genes were enriched in these signaling pathways linked to apoptosis, necroptosis, and immune response (Figure 7C). The PPI analysis revealed that the genes upregulated by Poly(I:C) and suppressed by SPD were enriched in four networks: Toll-like receptor and lymphocyte proliferation, DNA replication and activation of the pre-replicative complex, proteasome complex and ubiquitin homolog, and other networks (Figure 7D). Meanwhile, the genes downregulated by Poly(I:C) and induced by SPD were enriched in three networks related to the regulation of chemotaxis and vascular endothelial growth factor-rector, vascular smooth muscle contraction and blood pressure regulation, and 5–3 exoribonuclease activity (Figure 7E).

The transcriptomic analysis yielded results that were consistent with experimental observations, demonstrating that SPD effectively mitigates Poly(I:C)-induced inflammation, oxidative stress, and apoptosis in Leydig cells. This restoration of cellular function consequently enhances testosterone synthesis. Mechanistically, SPD appears to upregulate critical pathways involved in chemotaxis, vascular endothelial growth factor receptor signaling, vascular smooth muscle contraction, blood pressure regulation, and 5–3 exoribonuclease activity. Conversely, SPD downregulates several signaling pathways, including Toll-like receptor signaling, lymphocyte proliferation, DNA replication, and pre-replicative complex activation, as well as proteasome complex and ubiquitin homolog-mediated processes. These coordinated molecular mechanisms collectively contribute to SPD’s protective effects against Poly(I:C)-induced damage in Leydig cells.

## 3. Discussion

Previous studies have demonstrated that exogenous spermidine (SPD) not only does not induce oxidative stress, immune response, or cellular apoptosis, but also exhibits beneficial effects on growth promotion, intestinal health improvement, and antioxidant capacity enhancement in livestock and poultry [14,15,16,17]. Specifically, research on germ cells (GCs) has shown that SPD treatment maintains stable apoptosis rates without altering the expression of apoptosis-related genes, including Caspase8, Caspase9, and Bcl-2/Bax [18]. Furthermore, consistent findings have been reported regarding cell viability, with SPD showing no adverse effects on cellular survival [12]. Despite these well-documented benefits in non-ruminant species, the application of SPD in ruminants, particularly in the context of reproductive health under high-altitude conditions, remains largely unexplored and warrants further investigation.

To further evaluate the safety of SPD, we first conducted a toxicity experiment using a mouse model in which we examined whether long-term administration of SPD-containing water (10 mg/kg for 1, 7, and 14 days) would lead to liver dysfunction, abnormal hematological parameters, or histological changes in organ tissues. The results showed that SPD treatment did not cause significant liver damage, and no abnormalities were observed in hematological parameters or histological examination of organ tissues. These findings suggest that SPD exhibits good biological safety within the tested dosage range, providing supporting evidence for its potential use as a feed additive.

In this study, we first found that SPD alleviates Poly(I:C)-induced inflammatory immune response, apoptosis, and the decrease in testosterone secretion in Leydig cells of yaks. These results suggest that SPD can mitigate the adverse effects of Poly(I:C) on yak testicular Leydig cells and support the maintenance of their normal cellular functions. By activating cascade signaling pathways of TLR3 and NF-κB, Poly(I:C) enhances the expression of various immune cytokines and chemokines [19,20]. Previous studies indicated that Poly(I:C) has the ability to stimulate the production of immune cytokines like TNF-α and IL-16 [21]. This is in line with our findings, which revealed a significant increase in the expression of IL-6 and TNF-α in LCs stimulated by Poly(I:C). Our findings also confirmed that Poly(I:C) can cause oxidative stress and reduced testosterone production in LCs. Conceptually, inflammation can be viewed as a four-stage process, including a triggering system, a sensor mechanism, the transmission of the signal, the production of mediators, and the activation of cellular effectors, which may all affect oxidative stress or be affected, at various degrees, by oxidative stress [22]. Previous studies have shown that Poly(I:C) enhances the expression of IFN-β and apoptosis-related genes in cervical cancer cells by activating the NF-κB signaling pathway [23,24]. This activation leads to the release of cytochrome C from mitochondria into the cytoplasm and activates Caspase-9 and Caspase-3, stimulating both intrinsic and extrinsic apoptotic pathways and ultimately causing cell death [25,26]. Additionally, in endothelial cells, Poly(I:C) increases the expression levels of TNF and apoptosis-related ligands and their receptors, while downregulating the anti-apoptotic protein BCL-2, thereby activating both apoptotic signaling pathways [27]. In our studies, Poly(I:C) induced cell apoptosis, and also upregulated the expression of two NF-κB family subunits, P65 and REL, as well as MAPK and Caspase-3 and 6, which are involved in the mediation of cell apoptosis.

Numerous studies have shown that SPD plays a critical role in regulating immune cell functions [28]. Specifically, it promotes the polarization of macrophages toward an anti-immune phenotype, which aids in reducing inflammation [29,30,31]. Several recent studies have suggested that polyamines exert multiple effects including antioxidant and anti-immune benefits. Furthermore, in a pancreatitis model, polyamines could prevent damage to membrane structure caused by activated oxygen radicals by preventing the production of TNF-α and IL-6 production [32,33,34,35]. Our experiments demonstrated that SPD effectively alleviated Poly(I:C)-induced upregulation of immune factors in LCs, such as IL-8, IL-1α, TNF-α, RELA (P65), REL, and MAPK. We also verified that SPD significantly downregulated Poly(I:C)-induced upregulation of P65 and PP65 protein expression in LCs, which lays the foundation for further validation of whether SPD can rescue cellular inflammation through the NF-κB pathway.

SPD is recognized as a natural inducer of autophagy [10], which induces autophagy to exert an anti-apoptotic function. Moreover, in recent years there has been considerable evidence that SPD can mitigate cell apoptosis while maintaining cellular viability [36,37]. Some studies have focused on the issue that autophagy, apoptosis, and aging are connected in vivo to maintain the process of life in recent years [38]. In both in vitro and in vivo experiments, SPD was observed to promote autophagic flux and reduce apoptosis in cardiomyocytes [39,40]. The BCL-2 gene family, comprising proteins such as BAX and BCL-2, plays a vital role in regulating apoptosis by forming hetero- and homodimers in the mitochondrial membrane. The ratio of BCL-2 to BAX ultimately dictates the overall apoptotic effect [41,42]. In our study, SPD treatment led to elevated expression levels of the BCL-2 gene and protein, while simultaneously reducing BAX levels. This resulted in an increased BCL-2/BAX ratio. Our findings also indicated that SPD diminished the increase in Caspase-3 protein expression induced by Poly(I:C). Caspase activation marks the endpoint of the apoptosis signaling cascade, in which proCaspase-3 is converted to Caspase-3, a key biomarker and executor of cellular apoptosis. Caspase-3 subsequently catalyzes the hydrolysis of nucleic acids and cytoskeletal proteins [43,44]. In conclusion, SPD effectively mitigates Poly(I:C)-induced apoptosis.

The continuous generation of free radicals, originating from both internal and external sources, occurs as a result of normal metabolic activities within cells [45]. These free radicals can trigger oxidative stress, a damaging process that negatively affects all biomolecules, impairing cellular functions and potentially leading to cell death and various diseases [46,47]. In mammals, only a small fraction of the intracellular polyamine content comes from cellular biosynthesis and absorption by the gut microbiota [48], and it is obtained chiefly from the daily diet [49,50]. In particular, exogenous SPD supplementation through drinking water or food can activate autophagy, improve mitochondrial function, and exert antioxidative stress. SPD activated the Nrf2-Keap1-ARE antioxidant signaling pathway and peroxidase endogenous antioxidants such as CAT, MDA, and GSH to regulate the body’s antioxidant function [51]. Antioxidative enzymes such as SOD, CAT, and the antioxidant GSH act as the innate defense system against tissue damage from free radicals across various organs, including the testes [52,53]. Numerous studies have been conducted to show that SPD has a significant antioxidative stress effect [54]. Our experiments indicated that SPD significantly reduced the MDA increase and enhanced GSH, SOD, and CAT activities, which demonstrated that SPD could effectively alleviate Poly(I:C)-induced oxidative stress.

Many studies have demonstrated that testosterone synthesis in LCs is influenced by various autocrine and paracrine factors [47]. These factors encompass growth factors and cytokines, including TNFα, IL-1, and IL-6 [55]. This may indicate that the immune response directly affects testosterone production by LCs. Some enzymes involved in testosterone production include CYP11A1, CYP17A1, and StAR [56]. Interestingly, in our study, while Poly(I:C) did not lower the levels of CYP11A1 and CYP17A1, SPD significantly increased their expression. In contrast, Poly(I:C) markedly reduced StAR levels, but this effect was reversed by SPD. The observed upregulation of CYP11A1 and CYP17A1 in both the Poly(I:C) and SPD groups suggests a compensatory mechanism in response to inflammatory stress. Poly(I:C), a viral mimic, induces a robust inflammatory response, which may initially disrupt steroidogenesis. However, the upregulation of CYP11A1 and CYP17A1 could reflect an adaptive response by Leydig cells to maintain testosterone production despite the inflammatory challenge. This is consistent with previous studies showing that steroidogenic enzymes can be upregulated under stress conditions as part of a feedback mechanism to counteract the inhibitory effects of inflammation on steroidogenesis [57,58]. In the SPD group, the increase in CYP11A1 and CYP17A1 may be attributed to the anti-inflammatory and protective effects of SPD, which could mitigate the negative impact of inflammation on steroidogenic enzymes. SPD has been shown to enhance cellular resilience and promote the expression of key enzymes involved in steroidogenesis, thereby supporting testosterone synthesis even in the presence of inflammatory stimuli. These findings highlight the intricate balance between inflammation and steroidogenesis, where compensatory mechanisms may be activated to preserve testosterone production [59]. Further studies are needed to elucidate the precise molecular pathways involved in this regulation. We next analyzed testosterone levels in LCs and discovered that SPD could mitigate the decrease in testosterone caused by Poly(I:C). To our knowledge, no previous studies have examined whether SPD can restore the impaired function of male germ cells.

Our transcriptome analysis revealed that SPD modulates the expression of genes associated with immune responses, oxidative stress, and apoptosis. Specifically, SPD restored the expression of genes downregulated by Poly(I:C) and suppressed the upregulation of genes associated with inflammatory signaling pathways. This suggests that SPD exerts a regulatory effect on the immune response and oxidative stress pathways activated by Poly(I:C). The enrichment analysis of genes in pathways related to vascular development and differentiation indicates that SPD may promote cellular processes essential for maintaining Leydig cell function. Conversely, the suppression of genes involved in DNA replication and stress responses suggests a protective mechanism against Poly(I:C)-induced cellular damage.

While our study focused on the protective effects of SPD on Leydig cells under stress conditions, it is important to acknowledge the potential role of spermine (SPM) in modulating cellular responses. SPD and SPM are closely related polyamines, and SPD can be enzymatically converted to SPM in vivo by spermine synthase [60]. Although our experiments were conducted under controlled in vitro conditions, where the conversion of SPD to SPM may be limited, future studies should explore the interplay between SPD and SPM in Leydig cell function and testis development. Specifically, further research could investigate the extent of SPD-to-SPM conversion in vivo using inhibitors of spermine synthase or genetic models, compare the distinct roles of SPD and SPM in cellular physiology and stress responses, employ high-throughput transcriptomic and proteomic analyses to identify key pathways and molecular targets regulated by these polyamines, and evaluate the therapeutic potential of SPD and SPM in treating testicular dysfunction or infertility caused by oxidative stress or inflammation [14]. By addressing these questions, future studies could provide a more comprehensive understanding of the roles of SPD and SPM in Leydig cell biology and their potential applications in reproductive health [17].

In conclusion, SPD demonstrates a capacity to counteract Poly(I:C)-induced immune and oxidative challenges in yak Leydig cells, thereby preserving their function and viability. Further research is warranted to explore the therapeutic potential of SPD in male reproductive health, particularly in conditions associated with viral infections and oxidative stress. Exploring these upstream events could uncover additional layers of regulation and provide a more comprehensive understanding of its mechanism of action in future. The transcriptomic evidence underscores the potential of SPD to alleviate Poly(I:C)-induced inflammation and apoptosis in LCs. This study not only validates the anti-inflammatory and anti-apoptotic properties of SPD but also identifies new avenues for therapeutic applications.

Future research focusing on the interplay between SPD and its molecular targets, as well as in vivo studies, will be crucial for translating these findings into clinical settings. In addition, the metabolic differences of SPD among different animal species require further investigation [16]. Specifically, in high-altitude grazing yaks, the metabolic process, bioavailability, and long-term effects of SPD on reproductive health remain unclear. Future studies should focus on elucidating the specific mechanisms of SPD in ruminants and evaluating its feasibility in practical livestock production. This will provide a scientific basis for the potential application of SPD as a feed additive in economically important animals such as yaks.

## 4. Materials and Methods

### 4.1. Isolation and Identification of LCs

The testicles used in this study were obtained from three sexually mature male yaks (3–4 years old) raised at the Qingbaijiang Yak Farm in Chengdu, Sichuan Province, China. After slaughter, one testicle from each yak was transported to the Reproduction Laboratory at Sichuan Agricultural University for further processing and Leydig cell isolation.

Firstly, leucorrhea on the surface of the testis was removed to expose the parenchyma. After being washed 2–3 times, a few small tissues of approximately 1 mm^3^ were excised, and these tissues were washed with PBS until they appeared white. Secondly, these tissues were digested in Dulbecco’s Modified Eagle Media (DMEM/F-12) containing 200 IU/mL of collagenase IV at 37 °C on a shaker with 100 rpm for 30 to 40 min. Thirdly, the digested mixture was added equal volume of serum-free DMEM/F-12, and the supernatant containing a lot of isolated cells was filtered through a 70 µm cell strainer. Finally, the cell suspension was centrifuged at 1300 r/min for 5 min, the supernatant was discarded, and the dispersed cells were cultured in DMEM/F-12 with 10% FBS at 37 °C [52]. The cultured cells were observed under microscopy (IX-71, Olympus) (Appendix A).

### 4.2. Identification and Preservation of Testicular LCs

3β-hydroxysteroid dehydrogenase (3β-HSD) staining was conducted to assess cell purity. At least 10 fields were examined, counting 100 cells in each field. The total cell count was recorded, identifying the blue or dark blue stained cells as testicular LCs. The percent of LCs is counted by the formula: $\frac{\text{The number of blue or dark blue cells}}{\text{total number of cells}}$ × 100% [61] (Appendix A). The purity of LCs is more than 95%, and these cells are used for further experiments.

### 4.3. Experiment Design

(1) Firstly, to optimize the concentration and the induced time of Poly(I:C), Leydig cells were cultured in DMEM/Ham’s F12 medium containing 10% FBS and different concentrations of Poly(I:C) (0, 0.1, 0.5, and 1 μg/mL) for 6 h, and then Leydig cells were cultured in DMEM/Ham’s F12 medium containing 10% FBS and 0.5 μg/mL of Poly(I:C) for different time (0, 6, 12, and 24 h). These cells were collected for measuring the expression level of IL6 and TNF-α. The suitable concentration and induction time of Poly(I:C) for the Leydig cells are 0.5 μg/mL and 6 h. Secondly, Leydig cells were divided into three groups, the control, Poly(I:C), and SPD + Poly(I:C) groups. The control group was a blank group, cultured for 24 h. In the Poly(I:C) group, 0.5 μg/mL of Poly(I:C) was added for the last 6 h. In the SPD + Poly(I:C) group, 20 ng/mL of SPD was added at the beginning and 0.5 μg/mL of Poly(I:C) was added for the last 6 h. The Leydig cells were collected within 24 h to perform the next experimental analysis.

(2) To evaluate the in vivo biosafety of spermidine (SPD), male mice aged 5–8 weeks were obtained from Chengdu Dashuo Experimental Animal Co., Ltd. All mice were healthy and acclimatized for seven days before being randomly divided into two groups: one group received regular drinking water, while the other group was provided with drinking water supplemented with SPD at a dose of 10 mg/kg. During the 14-day feeding and observation period, the general health status of each mouse was monitored daily. At the end of the experiment, the mice were euthanized, and whole blood was collected for standard hematological analysis and serum biochemical testing. Additionally, major organs, including the heart, liver, spleen, lungs, and kidneys, were fixed in 4% paraformaldehyde for standard hematoxylin and eosin (H&E) staining. The animal study was conducted in accordance with protocols approved by the Animal Ethics Committee of Sichuan Agricultural University.

### 4.4. Safety Assessment of Spermidine

After administering drinking water containing 10 mg/kg spermidine to mice, blood routine tests were conducted on day 1, day 7, and day 14 to evaluate the following parameters: white blood cell count (including neutrophils, lymphocytes, monocytes, eosinophils, and basophils), hematological parameters (red blood cell count, hemoglobin, hematocrit, mean corpuscular volume, mean corpuscular hemoglobin, and mean corpuscular hemoglobin concentration), and liver function markers (alanine aminotransferase and aspartate aminotransferase). Additionally, pathological analysis was performed on the heart, liver, spleen, lungs, and kidneys. Tissues were sectioned, embedded, and stained with hematoxylin and eosin (H&E) for histopathological examination.

### 4.5. Cell Viability Analysis

The viability of cells was measured using the Cell Counting Kit-8 (CCK-8) assay. LCs were cultured in a 96-well plate for 24 h. A total of 10 μL of CCK-8 solution (Beyotime, Shanghai, China) was added to each well and then incubated for 2 h at 37 °C on a shaker. Finally, optical density (OD) at 450 nm was recorded using an optical reader (Varioskan™ LUX Thermo, Waltham, MA, USA).

### 4.6. Measurement of Cell Apoptosis

Apoptosis was assessed using the Annexin V-FITC/PI method (BD Bioscience, Franklin Lakes, NJ, USA). Each sample tube received 2.5 μL of PE and Annexin reagent, mixed thoroughly, and incubated in the dark at room temperature for 20 min. The apoptotic rate was analyzed using a FACSCalibur flow cytometer (BD Biosciences, NJ, USA).

### 4.7. RT-qPCR

The expression of genes related to immune response, apoptosis, and steroidogenesis was measured by RT-qPCR. RNA was extracted using a total RNA extraction kit (1 μg; Toyobo, Saka, Japan). The integrity and concentration of mRNA were assessed using an optical density (OD) reader at 260 and 280 nm wavelengths. The expression levels of these genes were measured using the SYBR Green qPCR kit (Takara, Dalian, China) on a real-time fluorescence quantitative PCR instrument (CFX-96, Bio-Rad, Cambridge, MA, USA), and then the program of melt curve was run for evaluating product specificity. All primer information is listed in Table 1.

### 4.8. ELISA for the Detection of SOD, CAT, GSH, MDA and Testosterone Contents in LCs

LCs in different groups were cultured in the T25 culture flask and the cells were harvested after being gently washed with cold PBS, then digested with trypsin and centrifuged at 1000 rpm for 5 min to collect the cells; the collected cells were washed 3 times with cold PBS. A total of 200 μL of PBS was added to each 1 × 10^6^ cells to re-suspend them in order to ensure consistent cell density in each group. The suspension was centrifuged at 1500 rpm for 10 min, and the cell samples required for the Elisa assay were prepared by ultrasonic crushing in an ice bath at a frequency of 20 kHz with a power of 150 W, repeated three times at intervals of 3 s for every 2 s of crushing. Subsequently, SOD (Cat. No.: RX1600268B), MDA (Cat. No.: RXJ1600848B), GSH (Cat. No.: JRX773476), CAT (Cat. No.: RX1600200B), and testosterone contents (Cat. No.: RXJ1600827B) in the cell samples were determined and calculated using an ELISA kit (Ruixinbio Quanzhou, China).

### 4.9. Western Immunoblotting

Total protein was extracted from ESCs using RIPA lysis buffer (Cat.No.: P0013B; Beyotime) containing PMSF (Cat.No.:ST505, Beyotime, Nanjing, China) and a phosphatase inhibitor cocktail (Cat.No.: P1045; Beyotime). Protein concentration was determined by a BCA protein assay kit (Cat.No.: P0012, Beyotime). Proteins were separated using sodium dodecyl sulfate-polyacrylamide gel electrophoresis (SDS-PAGE) and then transferred to nitrocellulose membranes. The membranes were blocked with 3% BSA for one hour at room temperature. Overnight incubation at 4 °C was performed with primary antibodies against nuclear factor kappa B (NF-κB) p65 (Cat. No.: L8F6, CST; 1:1000), phospho-p38 Mitogen-activated protein kinase (MAPK) (Cat. No.: 93H1, CST; 1:1000), IL-6 (Cat. No.: D3K2N, CST; 1:1000), Caspase-3 (Cat. No.: 9662, CST; 1:1000), and β-actin (Cat.No.: 8H10D10, CST; 1:1000). After being washed, the membranes were incubated with secondary antibodies (1:5000; goat anti-mouse IgG and goat anti-rabbit IgG; Absin) for one hour at room temperature. Finally, the protein bands were colored with sensitive chemiluminescence (Cat.No.: PK10002; Beyotime) then were digitally captured, and their intensity was analyzed using a GelDoc imaging system (GelDoc^TM^ XR; Bio-Rad, MA, USA).

### 4.10. RNA-Seq and Data Analysis

Total RNA was extracted and its concentration and purity were measured using a NanoDrop 2000 spectrophotometer (Thermo Fisher Scientific, Waltham, MA, USA), with an A260/A280 ratio between 1.8 and 2.0. RNA integrity was assessed using an Agilent 2100 Bioanalyzer, and samples with RIN values greater than 7 were used for sequencing. RNA sequencing was performed by Novogene (Beijing, China). cDNA libraries were constructed using the TruSeq RNA Sample Prep Kit v2 (Illumina, San Diego, CA, USA) and sequenced on an Illumina NovaSeq 6000 platform, generating paired-end reads of 150 bp in length. Data preprocessing and quality control were conducted using FastQC (v0.11.8), with low-quality reads and adapter sequences removed using Trim Galore (v0.6.6), and the mappe ratio was present in Appendix A. The raw data of all samples were stored in the NCBI SRA database (https://www.ncbi.nlm.nih.gov/bioproject/ (accessed on 13 March 2025): Accession No.: PRJNA1203441). Differentially expressed genes (DEGs) were identified using the DESeq2 package (v1.30.0) in R, with |log2(FC)| ≥ 1 and an adjusted *p*-value < 0.05. DEGs from the control, Poly(I:C), and Spd groups were compared, and a Venn diagram (Appendix A), Pearson’s correlation (Appendix A), and Heatmap (Appendix A) of DEGs were generated using the online platform provided by Novogene Cloud (https://www.novogene.com (accessed on 13 March 2025)). Trend analysis of the intersecting DEGs was conducted using the OmicShare platform (https://www.omicshare.com/ (accessed on 13 March 2025)), where genes were classified into distinct clusters based on their expression patterns (https://metascape.org (accessed on 13 March 2025)). Finally, these genes in clusters 1, 2, 3, and 6 were downregulated by Poly(I:C) and restored by SPD, while those genes in clusters 4, 5, 9, and 10 which were upregulated by Poly(I:C) and suppressed by SPD, were performed KEGG analysis, and constructed protein–protein interaction (PPI) networks using the STRING online tool (https://string-db.org (accessed on 13 March 2025)).

### 4.11. Statistical Analysis

Data analysis was performed using GraphPad Prism 9 was employed for plotting. Experimental results are presented as mean ± SD. For group comparisons, ANOVA followed by Tukey’s test was applied, with a significance threshold set at * *p* < 0.05, ** *p* < 0.01, *** *p* < 0.001 and **** *p* < 0.0001. All analyses were conducted using standardized workflows on respective platforms to ensure the reproducibility and reliability of the results.

## 5. Conclusions

Our study provides compelling evidence that spermidine has significant protective effects on yak Leydig cells exposed to Poly(I:C)-induced immune response and oxidative stress. By modulating proinflammatory cytokines, reducing apoptosis, enhancing antioxidant defenses, and restoring testosterone production, SPD demonstrates its potential as a therapeutic agent to safeguard male reproductive health. These findings suggest that SPD might be developed as a feed additive or dietary supplement to improve male fertility, particularly in livestock facing reproductive challenges induced by viral infections. Given the increasing concern over viral-induced infertility in farm animals and its broader implications for human health, further studies, particularly in vivo models, are warranted to fully explore the mechanisms of SPD action and its applicability in both agricultural and clinical settings.

## Figures and Tables

**Figure 1 ijms-26-02753-f001:**
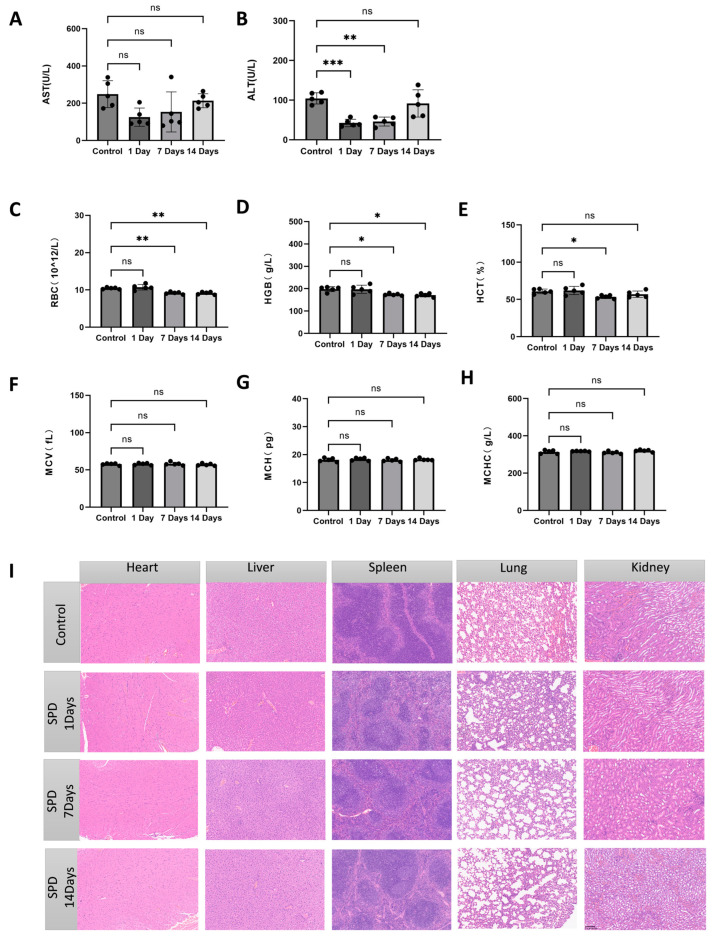
Effects of SPD on hematological and biochemical parameters as well as histological changes in major organs of mice. (**A**–**H**) Levels of AST, ALT, RBC, HGB, HCT, MCV, MCH, and MCHC in different treatment groups (control group and SPD-treated group) on days 1, 7, and 14. Data are presented as mean ± standard deviation, with five replicates per group. * *p* < 0.05, ** *p* < 0.01, *** *p* < 0.001, ns: no significant difference. (**I**) HE-stained histological sections of major organs (heart, liver, spleen, lung, and kidney) in each treatment group, showing the morphological characteristics of the tissues. Scale bar: 100 μm.

**Figure 2 ijms-26-02753-f002:**
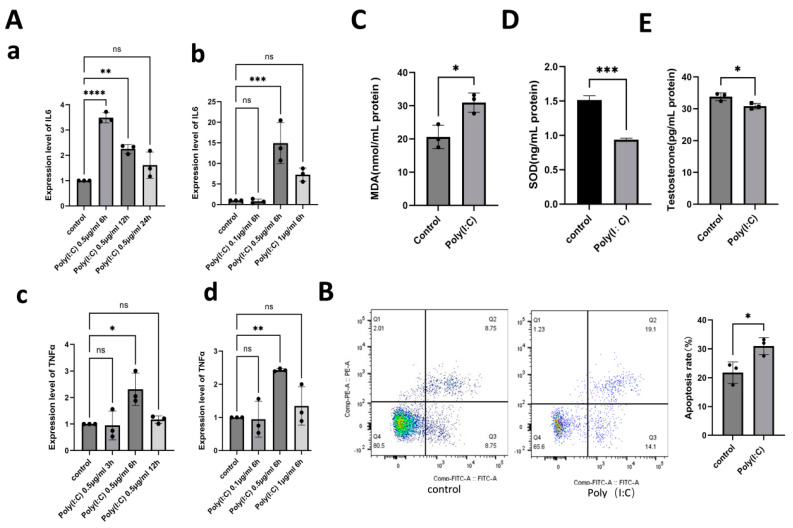
Poly(I:C) induces immune response, apoptosis, oxidative stress, and function damage of LCs. (**A**) The expression level of IL-6 (**Aa**,**Ab**) and TNF-α (**Ac**,**Ad**) in the LCs which were treated with different concentrations (0.1 μg/mL, 0.5 μg/mL, 1 μg/mL) of Poly(I:C) and at different times (6 h, 12 h, and 24 h for IL-6, and 3 h, 6 h, and 12 h for TNF-α). (**B**) Detection of apoptosis by flow cytometry. (**C**) The concentration of MDA in the Poly(I:C) group is significantly higher than that in the control. (**D**) The concentration of SOD in Poly(I:C) group is significantly lower than that in the control. (**E**) Intracellular testosterone content in Leydig cells significantly decreases after Poly(I:C)-induction. The data are averages of at least three independent experiments (mean ± SD, n = 3), ns: no significant difference, *p* > 0.05, * *p* < 0.05, ** *p* < 0.01, *** *p* < 0.001, and **** *p* < 0.0001.

**Figure 3 ijms-26-02753-f003:**
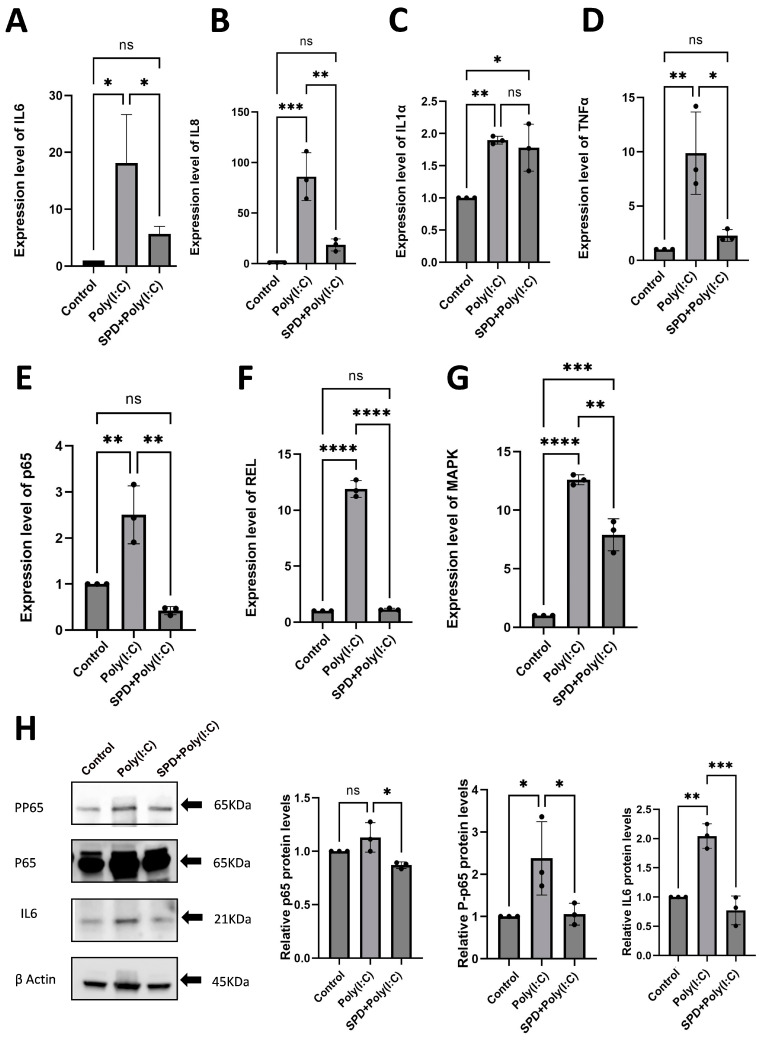
SPD attenuates Poly(I:C)-induced immune response. The mRNA levels of IL6 (**A**), IL8 (**B**), IL1α (**C**), TNFα (**D**), P65 (**E**), REL (**F**), MAPK (**G**) of LCs in the Poly(I:C) and SPD + Poly(I:C) groups. The protein expression levels of phosphorylation P65 (PP65), P65, and IL6 of LCs (**H**) in the Poly(I:C) and SPD + Poly(I:C) groups. The data are averages of at least three independent experiments (mean ± SD, n = 3), ns: no significant difference, *p* > 0.05, * *p* < 0.05, ** *p* < 0.01, *** *p* < 0.001 and **** *p* < 0.0001.

**Figure 4 ijms-26-02753-f004:**
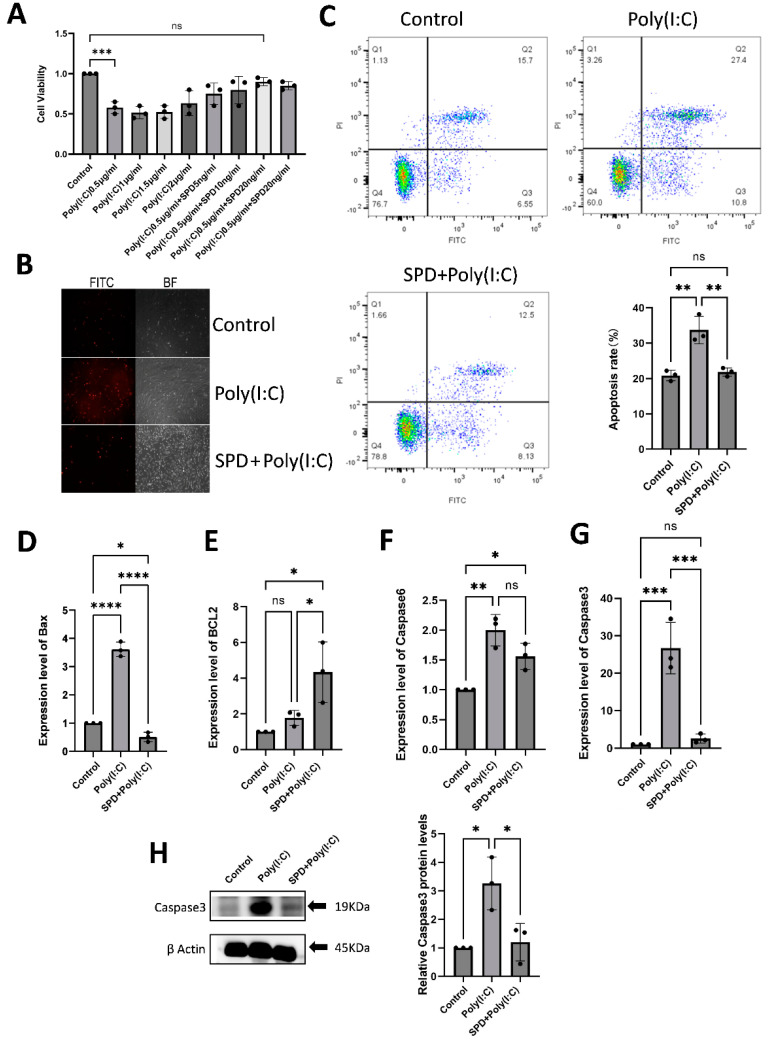
SPD recuses Poly(I:C)-induced apoptosis of LC. (**A**) Cell viability (**B**) Early apoptotic cell dye with Annexin V-FITC, BF stands for bright field. (**C**) Detection of apoptosis by flow cytometry. The expression levels of BAX (**D**), BCL-2 (**E**), Caspase-6 (**F**), and Caspase-3 (**G**) of LCs in the control, Poly(I:C) and SPD + Poly(I:C) group. (**H**) The protein expression levels of Caspase-3 of LCs in the control, Poly(I:C) and SPD + Poly(I:C) group. The data are averages of at least three independent experiments (mean ± SD, n = 3), ns: no significant difference, *p* > 0.05, * *p* < 0.05, ** *p* < 0.01, *** *p* < 0.001 and **** *p* < 0.0001.

**Figure 5 ijms-26-02753-f005:**
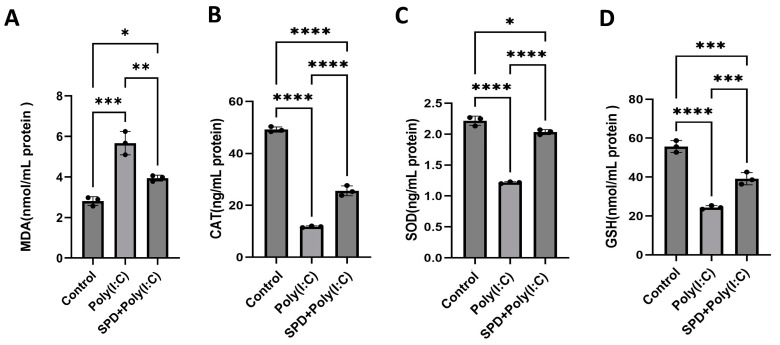
SPD alleviated Poly(I:C)-induced oxidative stress of LCs. (**A**) The concentration of MDA (**A**), CAT (**B**), SOD (**C**), and GSH (**D**) of LCs in the control, Poly(I:C), and SPD + Poly(I:C) group. The data are averages of at least three independent experiments (mean ± SD, n = 3). * *p* < 0.05, ** *p* < 0.01, *** *p* < 0.001, and **** *p* < 0.0001.

**Figure 6 ijms-26-02753-f006:**
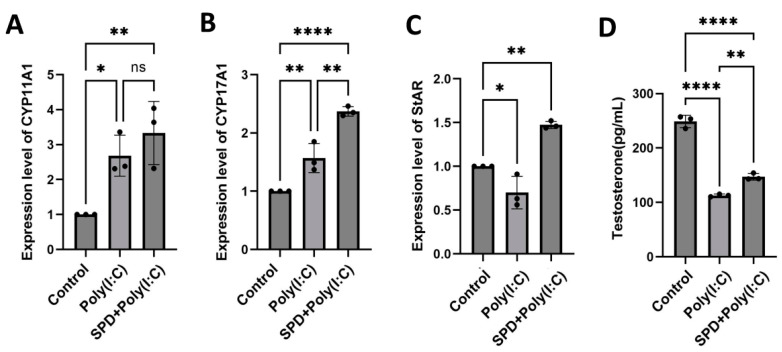
SPD alleviated the decrease in testosterone synthesis caused by Poly(I:C) in LCs. The expressive levels of CYP11A1 (**A**), CYP17A1 (**B**), and StAR (**C**) in LCs. The concentration of testosterone of LCs in the control, Poly(I:C), and SPD + Poly(I:C) group (**D**). The data are averages of at least three independent experiments (mean ± SD, n = 3). ns: no significant difference, *p* > 0.05, * *p* < 0.05, ** *p* < 0.01, and **** *p* < 0.0001.

**Figure 7 ijms-26-02753-f007:**
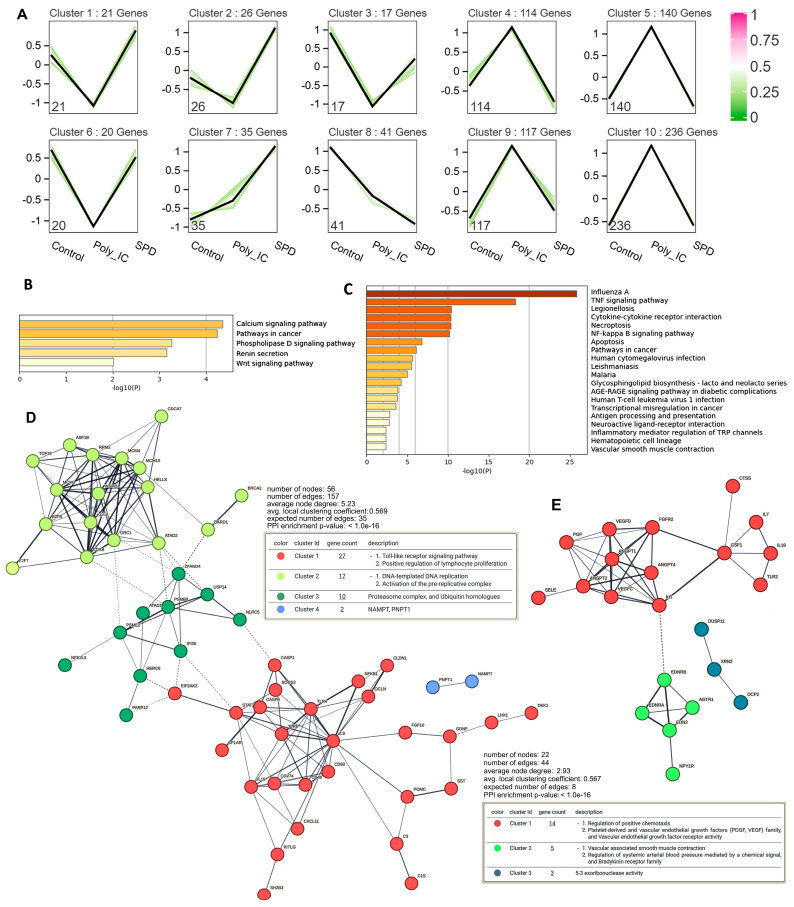
Transcriptome analysis reveals gene expression trends and pathway enrichment in control, Poly(I:C), and SPD+ Poly(I:C) groups. (**A**) Trend analysis of co-expressed differentially expressed genes (DEGs) based on their expression levels in the control group, Poly(I:C) group, and Poly(I:C) + SPD group (SPD means Poly(I:C) + SPD). Genes with similar expression patterns were grouped into clusters. Genes in clusters 1, 2, 3, and 6 were downregulated after Poly(I:C) treatment and restored by SPD, while genes in clusters 4, 5, 9, and 10 were upregulated after Poly(I:C) treatment and suppressed by SPD. KEGG pathway enrichment analysis of genes in clusters 1, 2, 3, and 6 (**B**) and in clusters 4, 5, 9, and 10 (**C**). PPI network analysis of genes in clusters 4, 5, 9, and 10 (**D**), and in clusters 1, 2, 3, and 6 (**E**). During PPI analysis, genes with node degree ≤ 2 were excluded, and K-means method was used for network clustering. The dashed lines represent predicted interactions, while solid lines indicate known interactions from curated databases.

**Table 1 ijms-26-02753-t001:** The primer information and annealing temperature for RT-qPCR.

Gene Symbol	Primer Sequence (5′-3′)	Annealing Temperature (°C)	Product Size (bp)	Accession Number
IL6	F: CTCGTATGCCAATGCCCTCA R: CCCAGATTGGAAGCATCCGT	60 °C	195	XM_005901249.2
TNFα	F: CTCGTATGCCAATGCCCTCA R: TGGTAGGAGACTGCAATGCG	60 °C	174	NM_173966.3
REL	F: GTAAAGATGCAGTTGCGGCG R: CTCCACAATCCTGCCACAGT	60 °C	151	XM_005889034.1
P65	F: GCCAGGTTCCAGACCTCTTC R: ATAGTGGGGTGGGTCTTGGT	60 °C	187	XM_005894097.1
IL8	F: ACCCCAAGGAAAAGTGGGTG R: CCCACACAGTACATGAGGCA	60 °C	183	XM_005891246.2
MAPK	F: TATTCGAGCACCGACCATCG R: GCAGCAGGTTGGAAGGTTTG	60 °C	203	NM_175793.2
CYP11A1	F: TTCAACCTCATCCTGACGCC R: GTGCAAGAGGTGTGGACTGA	60 °C	204	NM_176644.2
CYP17A1	F: GATCGTGGCCTACCTGCTAC R: CCACAACGTCTGTGCCTTTG	60 °C	242	NM_174304.3
StAR	F: CTGCCCTGCTCTTGAAGCTA R: GAAAACGTGCCACCACCTTG	60 °C	160	NM_174189.3
Caspase-6	F: GCTAAGCTCTCCGCTACGAT R: CCTGTTCGGCAGGGTTAAGT	60 °C	210	XM_005896232.2
Caspase-3	F: CGTCGTAGCTGAACCGTGA R: TTACTGCATCCTGTCTCCTCCT	60 °C	164	XM_010820245.4
BAX	F: CTCTGAGCAGATCATGAAGACAG R: CAGAAAACATTTCAGCCGCCA	60 °C	260	NM_173894.1
BCL-2	F: CGGAGCAGCCTGTTTAGGAA R: ACAAAAGCGGTTTCTCACGC	60 °C	127	XM_005224105.5

## Data Availability

All data of the study have been submitted as a Appendix A.

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
