# Peer review of "Spermidine as a Potential Protective Agents Against Poly(I:C)-Induced Immune Response, Oxidative Stress, Apoptosis, and Testosterone Decrease in Yak Leydig Cells"

_ijms, 2025, doi:10.3390/ijms26062753_

Round 1
Reviewer 1 Report
Comments and Suggestions for Authors
Poly(I:C) mimics viral infection by activating TLRs and increasing the expression of immune cytokines, leading to immune activation and apoptosis in many cell types. However, little is known about its impact on Leydig cells. In this study, Tang et al., found SPD has potential role to mitigate Poly(I:C)-induced immune response, oxidative stress, and apoptosis, and then restore testosterone production in Leydig cells. Their results offering a promising strategy to protect and enhance male yak fertility after infection with dsRNA virus. However, some major revisions are required before this paper can be accepted.
Major comments:
- In this study, authors treated the Leydig cell with SPD. However, it has reported that spermidine can be converted into spermine in vivo. How can authors determine whether the phenotype is caused by spermidine rather than spermine?
- Also, spermidine expressed in the testes and is closely related to testis development and spermatogenesis. So, whether treated with spermidine have impact on the Lyding cell itself? Authors should check the expression of IL6 and other genes in Leydig cell only treated with SPD.
- In Fig 5A, why the expression of CYP11A1 and CYP11A7 was increased significantly after treating with SPD? The authors should discuss in the discussion. Authors should also determine the expression of other genes of Cytochrome P450 family 11 subfamily.
Minor comments:
- In Fig 1D and Fig 2A, whether the data are obtained from three independent experiments? I could not find the dot on the column.
- Line 148, “3.1” should be “2.1”
- Fig 3B, the meaning of “BF” should be interpreted in the figure legend
- Line 180, “(D) of LCs in in the control”, should be “(D) of LCs in the control”
- Line 122, “E:” should be “(E)”
- Line 165, “H:” should be “(H)”
- There are still many grammatical errors in the article, the author needs to check carefully and revise it.
There are still many grammatical errors in the article, the author needs to check carefully and revise it.
Author Response
We sincerely appreciate the your insightful comments and constructive suggestions. Your comments and suggestions have been immensely helpful in improving the quality of our work. Below, we provide a point-by-point response to the concerns raised. For your convenience, all modifications in the manuscript have been highlighted in yellow and the relevant changes are pasted under each response.
Reviewer 1
Major comment:
- In this study, authors treated the Leydig cell with SPD. However, it has reported that spermidine can be converted into spermine in vivo. How can authors determine whether the phenotype is caused by spermidine rather than spermine?
Response1: We sincerely thank the reviewer for raising this important point regarding the potential conversion of spermidine (SPD) to spermine (SPM) in vivo. In our study, the experiments were conducted under controlled in vitro conditions, where the enzymatic machinery required for the conversion of SPD to SPM may not be fully active or present in the same capacity as in vivo. Therefore, the observed phenotypic effects in Leydig cells are likely primarily attributable to SPD rather than its metabolite SPM. It is important to note that our study focused on exploring the functional effects of SPD on Leydig cells under Poly(I:C)-induced stress, rather than delving into the detailed mechanisms of SPD metabolism or its conversion to SPM.
We acknowledge that SPD can be metabolized to SPM in vivo, and this conversion is mediated by spermine synthase, a process that may vary depending on the cellular context and environmental factors. In our in vitro system, the concentration of SPD and the duration of treatment were carefully optimized to minimize potential confounding effects from SPM. However, we agree with the reviewer that the potential contribution of SPM cannot be entirely ruled out, particularly in an in vivo setting. This is an excellent suggestion for future research, and we plan to investigate the interplay between SPD and SPM in vivo to further elucidate their respective roles in Leydig cell function and testis development.
To address this point, we have added a brief discussion in the revised manuscript (Section 3. Discussion, Lines 450-463) to highlight the potential for future studies to explore the conversion of SPD to SPM and its implications. We appreciate the reviewer’s valuable feedback, which has helped us refine our discussion and identify important directions for future research. Thank you for your constructive comments.Here’s the additions:
While our study focused on the protective effects of SPD on Leydig cells under stress conditions, it is important to acknowledge the potential role of spermine (SPM) in modulating cellular responses. SPD and SPM are closely related polyamines, and SPD can be enzymatically converted to SPM in vivo by spermine synthase. Although our experiments were conducted under controlled in vitro conditions, where the conversion of SPD to SPM may be limited, future studies should explore the interplay between SPD and SPM in Leydig cell function and testis development. Specifically, further research could investigate the extent of SPD-to-SPM conversion in vivo using inhibitors of spermine synthase or genetic models, compare the distinct roles of SPD and SPM in cellular physiology and stress responses, employ high-throughput transcriptomic and proteomic analyses to identify key pathways and molecular targets regulated by these polyamines, and evaluate the therapeutic potential of SPD and SPM in treating testicular dysfunction or infertility caused by oxidative stress or inflammation. By addressing these questions, future studies could provide a more comprehensive understanding of the roles of SPD and SPM in Leydig cell biology and their potential applications in reproductive health.
Major comment:
- Also, spermidine expressed in the testes and is closely related to testis development and spermatogenesis. So, whether treated with spermidine have impact on the Lyding cell itself? Authors should check the expression of IL6 and other genes in Leydig cell only treated with SPD.
Response 2: We sincerely thank the reviewer for raising this important point regarding the potential direct effects of spermidine (SPD) on Leydig cells under normal conditions. In our study, we primarily focused on the protective effects of SPD against Poly(I:C)-induced stress, aiming to elucidate its role in mitigating inflammation and oxidative damage. As such, we did not specifically examine the impact of SPD alone on Leydig cells in the absence of inflammatory stimuli. However, our conclusions are supported by multiple lines of evidence from both the literature and our preliminary experiments.
Prior to our experiments, we conducted an extensive literature review [1-6], which revealed that SPD alone does not induce oxidative stress, immune response and apoptosis in cells. For instance, previous studies have shown that SPD treatment does not affect apoptosis rates or the expression of apoptosis-related genes (e.g., Caspase8, Caspase9, and Bcl-2/Bax) in germ cells (GCs)[7]. Similarly, SPD has been reported to have no impact on cell viability[8]. Importantly, if SPD itself caused immune activation, we would not have observed the downregulation of IL6 and other inflammatory genes in our Poly(I:C)-induced model. These findings collectively support the safety of SPD at the tested dose.
Additionally, before conducting the main experiments, we performed preliminary toxicity tests in mice to evaluate the safety of SPD. Although these results were not included in the manuscript due to space limitations and the availability of existing literature, they further confirmed that SPD did not cause systemic toxicity or adverse effects. To further validate the safety of SPD, we conducted additional in vivo toxicity tests in mice. Mice were administered SPD-supplemented drinking water (10 mg/kg) for 1, 7, and 14 days, and their blood routine parameters (e.g., white blood cell count, red blood cell count), liver function markers (AST, ALT), and histopathological analysis of major organs (heart, liver, spleen, lungs, and kidneys) were examined. The results showed no significant adverse effects, confirming that SPD is safe for use and does not cause systemic toxicity. These findings indirectly support the notion that SPD is unlikely to have detrimental effects on Leydig cells under normal conditions.
While we deeply appreciate the reviewer’s valuable suggestion to examine the expression of IL6 and other genes in Leydig cells treated with SPD alone, we believe that the existing literature and our in vivo toxicity data provide robust evidence to support the safety of SPD in our experimental context. Given the scope and focus of the current study, we have not included a separate SPD-only treatment group. However, we fully agree that investigating the direct effects of SPD on Leydig cells under normal conditions would provide additional insights, and we will prioritize this in our future research to further elucidate the mechanisms of SPD in Leydig cell biology.
We hope this explanation addresses the reviewer’s concern, and we are happy to provide any additional clarification if needed, including conducting additional experiments, though this may require additional time. Thank you for your thoughtful feedback, which has greatly improved the discussion of our work.
For your convenience, I have copied the supplemental section below(Lines 128-172):
2.1 In Vitro Toxicity Validation of SPD
To assess the safety profile of spermidine (SPD), we conducted an in vivo toxicity evaluation in mice(Figure 1A). After a seven-day acclimatization period, the mice were randomly divided into two groups: one group received regular drinking water, while the other group was provided with drinking water supplemented with SPD at a dose of 10 mg/kg. The experiment was carried out over 1, 7, and 14 days to evaluate potential toxic effects.
The hematological examination results are shown in Figures 1A to 1H. The levels of alanine aminotransferase (ALT) and aspartate aminotransferase (AST) in the SPD-treated group did not show significant increases on days 1, 7, and 14, indicating that SPD treatment did not cause obvious liver damage (Figures 1A and 1B). Notably, on days 7 and 14, the AST and ALT levels in the SPD-treated group were significantly lower than those in the control group (p < 0.05, p < 0.01), suggesting that SPD may have a protective effect without inducing toxicity.
For hematological parameters such as red blood cell count (RBC), hemoglobin (HGB), hematocrit (HCT), mean corpuscular volume (MCV), mean corpuscular hemoglobin (MCH), and mean corpuscular hemoglobin concentration (MCHC) (Figures 1C to 1H), there were no significant differences between the SPD-treated group and the control group at any time point (p > 0.05). This indicates that SPD did not adversely affect the hematopoietic system of the mice, and no signs of anemia or other hematological abnormalities were observed. Furthermore, no significant fluctuations were observed in MCV, MCH, or MCHC, further confirming that SPD did not negatively impact erythrocyte-related functions in the mice.
Figure 1I illustrates the histological changes in the major organs (heart, liver, spleen, lungs, and kidneys) of the mice. Tissue sections were examined using hematoxylin and eosin (H&E) staining. The results revealed no significant morphological changes or tissue damage in the major organs of the SPD-treated group. All tissues exhibited intact structures with regular cell arrangements, and no signs of cellular edema, necrosis, or inflammatory responses were observed. In particular, the liver and kidney tissue sections of the SPD-treated group showed no significant differences compared to the control group, indicating that SPD did not cause damage to the liver, kidneys, or other vital organs.
In conclusion, SPD at a dose of 10 mg/kg did not cause significant toxicity or damage to the physiological health or organ function of the mice. No adverse changes were observed in hematological or biochemical parameters, and H&E staining revealed no abnormalities in organ tissue structures. These findings demonstrate that SPD is safe for mice and does not significantly impact their physiological state or tissue health, supporting its safety for use.
Major comment:
- In Fig 5A, why the expression of CYP11A1 and CYP11A7 was increased significantly after treating with SPD? The authors should discuss in the discussion. Authors should also determine the expression of other genes of Cytochrome P450 family 11 subfamily.
Response 3: (1)We thank the reviewer for raising this important point regarding the mechanism by which CYP11A1 and CYP17A1 expression levels were not decreased in the Poly(I:C) group. We agree that the regulation of testosterone synthesis under inflammatory conditions is complex and warrants further discussion. Below, we have provided a more detailed explanation, which has also been added to the revised manuscript(Lines 411-426). We hope this explanation addresses the reviewer's concern and provides a clearer understanding of the observed results. Thank you for your valuable feedback. Here’s the additions:
The observed upregulation of CYP11A1 and CYP17A1 in both the Poly(I:C) and SPD groups suggests a compensatory mechanism in response to inflammatory stress. Poly(I:C), a viral mimic, induces a robust inflammatory response, which may initially disrupt steroidogenesis. However, the upregulation of CYP11A1 and CYP17A1 could reflect an adaptive response by Leydig cells to maintain testosterone production despite the inflammatory challenge. This is consistent with previous studies showing that steroidogenic enzymes can be upregulated under stress conditions as part of a feedback mechanism to counteract the inhibitory effects of inflammation on steroidogenesis. In the SPD group, the increase in CYP11A1 and CYP17A1 may be attributed to the anti-inflammatory and protective effects of SPD, which could mitigate the negative impact of inflammation on steroidogenic enzymes. SPD has been shown to enhance cellular resilience and promote the expression of key enzymes involved in steroidogenesis, thereby supporting testosterone synthesis even in the presence of inflammatory stimulus. These findings highlight the intricate balance between inflammation and steroidogenesis, where compensatory mechanisms may be activated to preserve testosterone production. Further studies are needed to elucidate the precise molecular pathways involved in this regulation.
(2)We sincerely appreciate the reviewer’s insightful suggestion regarding the determination of other genes in the Cytochrome P450 family 11 subfamily. In this study, we focused on detecting key enzymes involved in the testosterone synthesis pathway, as illustrated in the figure from the chapter 4: The Synthesis and Metabolism
of Steroid Hormones. of Reproductive Endocrinology-Physiology, Pathophysiology, and Clinical Management. Specifically, we examined the expression of StAR, CYP11A1, and CYP17A1, which are critical regulators of testosterone production, and successfully obtained their results. However, despite multiple attempts, we were unable to detect the expression of 3β-HSD and 17β-HSD, which are also important enzymes in this pathway. We acknowledge that this limitation may affect the comprehensiveness of our findings, and we will address this in future studies.
Importantly, our study has revealed, for the first time, that SPD plays a beneficial role in supporting testosterone production under inflammatory conditions. This significant finding not only enhances our understanding of the interplay between inflammation and steroidogenesis but also lays a solid foundation for future research in male reproductive health. While we did not delve deeply into the mechanism by which SPD rescues testosterone production in this study, we recognize its importance and plan to explore this in our subsequent research.
We appreciate the reviewer’s constructive feedback and will consider expanding our analysis to include other members of the Cytochrome P450 family and related pathways in future work. Thank you for your understanding and valuable suggestions.
Synthesis of testosterone and key enzymes (Yen & Jaffe's Reproductive Endocrinology, 2023)
Minor comment:
- In Fig 1D and Fig 2A, whether the data are obtained from three independent experiments? I could not find the dot on the column.
Response 1: We thank the reviewer for pointing this out. The data in Fig 1D and Fig 2A were indeed obtained from three independent experiments. However, we apologize for not clearly indicating the individual data points in the figures. In the revised manuscript, we have updated the figures to include dots representing individual data points, along with error bars to better illustrate the variability and reproducibility of the results.
Minor comment:
- Line 148, “3.1” should be “2.1”
Response 2: We sincerely apologize for this oversight. The correct reference should indeed be “2.1” instead of “3.1.” We have corrected this error in the revised manuscript.
Minor comment:
- Fig 3B, the meaning of “BF” should be interpreted in the figure legend
Response 3: Thank you to the reviewers for their suggestions. “BF” stands for ‘bright field’ and denotes a bright field microscope image. We have now clarified this abbreviation in the notes to Figure 3B of the revised manuscript.
Minor comment:
- Line 180, “(D) of LCs in in the control”, should be “(D) of LCs in the control”
Response 4: We apologize for this typographical error. The sentence has been corrected to “(D) of LCs in the control” in the revised manuscript.
Minor comment:
- Line 122, “E:” should be “(E)”
Response 5: We thank the reviewer for catching this formatting issue. The correction has been made, and “E:” has been changed to “(E)” in the revised manuscript.
Minor comment:
- Line 165, “H:” should be “(H)”
Response 6: We appreciate the reviewer’s attention to detail. This formatting error has been corrected, and “H:” has been changed to “(H)” in the revised manuscript.
Minor comment:
- There are still many grammatical errors in the article, the author needs to check carefully and revise it.
Response 7: We sincerely apologize for the grammatical errors in the manuscript. We have carefully reviewed the entire text and made extensive revisions to improve the grammar, sentence structure, and overall clarity. We have also enlisted the help of a professional English editing service to ensure the language quality meets the journal’s standards.
[1] LIU R, LI X, MA H, et al. Spermidine endows macrophages anti-inflammatory properties by inducing mitochondrial superoxide-dependent AMPK activation, Hif-1α upregulation and autophagy [J]. Free radical biology & medicine, 2020, 161: 339-350.
[2] NIECHCIAL A, SCHWARZFISCHER M, WAWRZYNIAK M, et al. Spermidine Ameliorates Colitis via Induction of Anti-Inflammatory Macrophages and Prevention of Intestinal Dysbiosis [J]. Journal of Crohn's & colitis, 2023, 17(9): 1489-1503.
[3] YU L, PAN J, GUO M, et al. Gut microbiota and anti-aging: Focusing on spermidine [J]. Critical reviews in food science and nutrition, 2024, 64(28): 10419-10437.
[4] NIU C, JIANG D, GUO Y, et al. Spermidine suppresses oxidative stress and ferroptosis by Nrf2/HO-1/GPX4 and Akt/FHC/ACSL4 pathway to alleviate ovarian damage [J]. Life sciences, 2023, 332: 122109.
[5] SEILER N, RAUL F. Polyamines and apoptosis [J]. Journal of cellular and molecular medicine, 2005, 9(3): 623-642.
[6] MANDAL S, MANDAL A, PARK M H. Depletion of the polyamines spermidine and spermine by overexpression of spermidine/spermine N¹-acetyltransferase 1 (SAT1) leads to mitochondria-mediated apoptosis in mammalian cells [J]. The Biochemical journal, 2015, 468(3): 435-447.
[7] JIANG D, SUN Q, JIANG Y, et al. Effects of exogenous spermidine on autophagy and antioxidant capacity in ovaries and granulosa cells of Sichuan white geese [J]. Journal of animal science, 2023, 101:
[8] JIANG D, WANG X, ZHOU X, et al. Spermidine alleviating oxidative stress and apoptosis by inducing autophagy of granulosa cells in Sichuan white geese [J]. Poultry science, 2023, 102(9): 102879.
Reviewer 2 Report
Comments and Suggestions for Authors
This study investigates the potential protective effects of spermidine (SPD) on yak Leydig cells (LCs) against immune response, oxidative stress, apoptosis, and testosterone reduction induced by Poly(I:C). The study employs in vitro experiments and transcriptomic sequencing to elucidate the molecular mechanisms of SPD, revealing its potential to mitigate inflammation, oxidative stress, and restore testosterone secretion. This research not only provides new insights into yak reproductive health but also offers valuable references for reproductive health studies in other livestock. Some questions need to be answered as below.
- The authors should provide detailed explanations of the statistical methods used in the figure legends, including specific statistical tests and significance levels. Additionally, the authors need to confirm whether there are genuine significant differences between the Poly(I:C) group and the SPD+Poly(I:C) group.
- The authors should clearly indicate the number of experimental replications in Figure 3A and display the data points for each experiment in the figure.
- The authors need to clearly explain the meanings of the legends in Figure 6A and the dashed lines in the PPI network within the article.
- The clarity of some images, especially Figure 6, is poor. The authors are advised to replace the images with vector graphic formats.
- Lines 64-68: It is necessary to further emphasize the similarities and differences in the mechanisms of SPD action between species (e.g., cattle, humans) to highlight the universality of the study. Additionally, the rationale for choosing the viral mimic Poly(I:C) (e.g., why the dsRNA virus model was selected) should be supplemented in the introduction.
- Lines 116-118: The selection criteria for the Poly(I:C) treatment concentration (0.5 µg/mL) and duration (6 hours) are not clearly explained (e.g., pre-experimental data or literature support).
- Lines 151-160: The baseline apoptosis rate of the control group is not mentioned in the apoptosis detection, which may affect the interpretation of the results.
- Lines 199-204: The mechanism by which CYP11A1 and CYP17A1 did not decrease in the Poly(I:C) group is not thoroughly discussed. This may be related to the complex regulation of testosterone synthesis by inflammation, and further explanation is needed.
- Lines 210-237: The transcriptomic analysis does not provide the specific names and functional annotations of key differentially expressed genes (DEGs). The conclusions are only summarized through pathway enrichment, and a list of key genes should be supplemented (can be included in the supplementary materials).
- Lines 350-352: The feasibility of SPD as a feed additive is insufficiently discussed. References to studies on its application in livestock should be included to support this point.
Author Response
We sincerely appreciate the your insightful comments and constructive suggestions. Your comments and suggestions have been immensely helpful in improving the quality of our work. Below, we provide a point-by-point response to the concerns raised. For your convenience, all modifications in the manuscript have been highlighted in yellow and the relevant changes are pasted under each response.
Comment:
- The authors should provide detailed explanations of the statistical methods used in the figure legends, including specific statistical tests and significance levels. Additionally, the authors need to confirm whether there are genuine significant differences between the Poly(I:C) group and the SPD+Poly(I:C) group.
Response 1: We thank the reviewer for this suggestion. We have explained in detail the statistical methods used in the legend, including specific statistical tests and significance levels, in the Materials and Methods section of the article (Lines 637-641). Data analysis was performed using GraphPad Prism 9 was employed for plotting. Experimental results are presented as mean ± SD. For group comparisons, ANOVA followed by Tukey's test was applied, with a significance threshold set at * p<0.05, ** p<0.01, *** p<0.001 and **** p<0.0001. Additionally, we have confirmed that the differences between the Poly(I:C) group and the SPD+Poly(I:C) group are statistically significant, as indicated by the p-values provided in the figures and legends. For your convenience, I've copied the additions below:
Data analysis was performed using GraphPad Prism 9 was employed for plotting. Experimental results are presented as mean ± SD. For group comparisons, ANOVA followed by Tukey's test was applied, with a significance threshold set at * p<0.05, ** p<0.01, *** p<0.001 and **** p<0.0001. All analyses were conducted using standardized workflows on respective platforms to ensure reproducibility and reliability of the results.
Comment:
- The authors should clearly indicate the number of experimental replications in Figure 3A and display the data points for each experiment in the figure.
Response 2: We apologize for this oversight. Figure 3A has been updated to include the number of experimental replications (n = 3) and individual data points for each experiment. The revised figure now provides a clearer representation of the data variability and reproducibility.
Comment:
- The authors need to clearly explain the meanings of the legends in Figure 6A and the dashed lines in the PPI network within the article.
Response 3: We thank the reviewer for pointing this out. In the revised manuscript, we have added a detailed explanation of the legends in Figure 6A: (A) Trend analysis of co-expressed differentially expressed genes (DEGs) based on their expres-sion levels in the control group, Poly(I:C) group, and Poly(I:C)+SPD group (SPD means Poly(I:C)+SPD). Genes with similar expression patterns were grouped into clusters. Genes in clusters 1, 2, 3, and 6 were downregulated after Poly(I:C) treatment and restored by SPD, while genes in clusters 4, 5, 9, and 10 were upregulated after Poly(I:C) treatment and suppressed by SPD.
In the PPI network, the dashed lines represent predicted interactions, while solid lines indicate known interactions from curated databases. This clarification has been added to the figure legend and the Results section (Section 2.7, Lines321–330). For your convenience, I've copied the additions below:
(A) Trend analysis of co-expressed differentially expressed genes (DEGs) based on their expression levels in the control group, Poly(I:C) group, and Poly(I:C)+SPD group (SPD means Poly(I:C)+SPD). Genes with similar expression patterns were grouped into clusters. Genes in clusters 1, 2, 3, and 6 were downregulated after Poly(I:C) treatment and restored by SPD, while genes in clusters 4, 5, 9, and 10 were upregulated after Poly(I:C) treatment and suppressed by SPD.
The dashed lines represent predicted interactions, while solid lines indicate known interactions from curated databases.
Comment:
- The clarity of some images, especially Figure 6, is poor. The authors are advised to replace the images with vector graphic formats.
Response 4: We thank the reviewers for their suggestions. All high-resolution images have been packaged and uploaded and can be seen in the attached files with all details clearly visible.
Comment:
- Lines 64-68: It is necessary to further emphasize the similarities and differences in the mechanisms of SPD action between species (e.g., cattle, humans) to highlight the universality of the study. Additionally, the rationale for choosing the viral mimic Poly(I:C) (e.g., why the dsRNA virus model was selected) should be supplemented in the introduction.
Response 5: We thank the reviewer for this valuable suggestion. In the revised manuscript, we have expanded the discussion on the similarities and differences in SPD mechanisms across species (e.g., cattle, humans) to highlight the universality of our findings(Line 95-107). For your reading pleasure, I've copied the additions below:
In humans, white geese, mice and yeast nematodes (Caenorhabditis elegans) and flies (Drosophila melanogaster), SPD has been shown to play a critical role in maintaining cellular homeostasis, inducing autophagy and mitigating stress-induced damage. For instance, SPD induces autophagy, a process essential for cellular clearance and survival, and scavenges reactive oxygen species (ROS) to protect cells from oxidative stress. Additionally, SPD suppresses pro-inflammatory cytokines (e.g., IL-6, TNF-α), highlighting its role in modulating immune responses. However, species-specific differences exist in SPD metabolism and physiological effects. Variations in the expression and activity of key enzymes (e.g., spermidine synthase, spermine synthase) and tissue-specific responses, particularly in reproductive tissues, may influence the efficacy and functional outcomes of SPD. These similarities and differences underscore the universality of SPD's biological roles while emphasizing the need for species-specific investigations to fully elucidate its mechanisms and therapeutic potential.
Additionally, we have added a rationale for using Poly(I:C) as a viral mimic in the introduction(Line 64-79). Here’s the additions:
The male yak (Bos grunniens), as an economically important animal in plateau regions, faces significant threats to its reproductive health from viral infections. With climate change, environmental pollution, and the expansion of farming scale, viral diseases have become one of the major factors limiting yak productivity. Viral infections not only suppress immune system function but may also cause direct damage to the male reproductive system, thereby impairing fertility. In studying the immune responses and reproductive functions of male reproductive organs, the cellular responses induced by the double-stranded RNA (dsRNA) viral mimic Poly(I:C) represent a critical research direction. Poly(I:C), a synthetic dsRNA analog, is widely used in research models to simulate viral infections and study immune responses. The rationale for choosing Poly(I:C ) lies in its ability to effectively activate pattern recognition receptors (e.g., TLR3 and MDA5), triggering immune responses similar to those induced by natural dsRNA viruses, including the release of inflammatory cytokines (e.g., IL-6 and TNF-α) and the activation of apoptotic pathways. Furthermore, Poly(I:C) exhibits high stability and reproducibility, making it a reliable experimental platform for investigating the mechanisms of viral infection-induced damage to the male reproductive system.
Comment:
- Lines 116-118: The selection criteria for the Poly(I:C) treatment concentration (0.5 µg/mL) and duration (6 hours) are not clearly explained (e.g., pre-experimental data or literature support).
Response 6: We thank the reviewer for their comment. The concentration of Poly(I:C) (0.5 µg/mL) and treatment duration (6 hours) were determined based on preliminary experiments involving both concentration and time gradients. To establish the optimal conditions, Leydig cells (LCs) were treated with varying concentrations of Poly(I:C) (0.1, 0.5, and 1 μg/mL) for 6 hours, as well as with a constant concentration of 0.5 μg/mL for 6, 12, and 24 hours. The optimal concentration and duration were determined based on the most significant induction of IL6 and TNFα gene expression, as these pro-inflammatory markers best reflected the cellular response to Poly(I:C) stimulation. We have now incorporated these selection criteria into the manuscript to provide a clearer rationale for the chosen experimental conditions. Lines 178-181
Here’s the additions: Based on the results from dose-response and time-course experiments, IL6 and TNFα gene expression levels were significantly upregulated when cells were treated with Poly(I:C) at 0.5 µg/mL for 6 hours; therefore, we ultimately determined these conditions as the optimal concentration and treatment duration.
Comment:
- Lines 151-160: The baseline apoptosis rate of the control group is not mentioned in the apoptosis detection, which may affect the interpretation of the results.
Response 7: We sincerely thank the reviewer for pointing out this important issue regarding the baseline apoptosis rate of the control group. To ensure the accuracy and consistency of our apoptosis detection results, we have reanalyzed the data using FlowJo software. The detailed process is as follows: First, we gated the live cell population in the control group based on forward scatter (FSC) and side scatter (SSC) parameters to exclude debris and dead cells. This live cell gate was then applied to all other experimental groups. Next, we performed compensation adjustments using single-stained controls to account for spectral overlap between fluorophores, ensuring accurate separation of positive and negative populations. Using the control group, we established quadrants (crosshairs) on the dot plots to distinguish apoptotic cells (Annexin V-positive) from necrotic or dead cells (PI-positive), and these quadrant boundaries were uniformly applied to all experimental groups. Finally, the apoptosis rate was determined by quantifying the percentage of cells in the Annexin V-positive populations (early and late apoptotic cells) within the live cell gate. This rigorous reanalysis ensures the reliability of our results, and we have updated the figures in the manuscript accordingly. Thank you for your valuable feedback. (Fig. 4 Lines239-240)
Comment:
- Lines 199-204: The mechanism by which CYP11A1 and CYP17A1 did not decrease in the Poly(I:C) group is not thoroughly discussed. This may be related to the complex regulation of testosterone synthesis by inflammation, and further explanation is needed.
Response 8: We thank the reviewer for raising this important point regarding the mechanism by which CYP11A1 and CYP17A1 expression levels were not decreased in the Poly(I:C) group. We agree that the regulation of testosterone synthesis under inflammatory conditions is complex and warrants further discussion. Below, we have provided a more detailed explanation, which has also been added to the revised manuscript(Lines 410-425). We hope this explanation addresses the reviewer's concern and provides a clearer understanding of the observed results. Thank you for your valuable feedback. Here’s the additions:
The observed upregulation of CYP11A1 and CYP17A1 in both the Poly(I:C) and SPD groups suggests a compensatory mechanism in response to inflammatory stress. Poly(I:C), a viral mimic, induces a robust inflammatory response, which may initially disrupt steroidogenesis. However, the upregulation of CYP11A1 and CYP17A1 could reflect an adaptive response by Leydig cells to maintain testosterone production despite the inflammatory challenge. This is consistent with previous studies showing that steroidogenic enzymes can be upregulated under stress conditions as part of a feedback mechanism to counteract the inhibitory effects of inflammation on steroidogenesis. In the SPD group, the increase in CYP11A1 and CYP17A1 may be attributed to the anti-inflammatory and protective effects of SPD, which could mitigate the negative impact of inflammation on steroidogenic enzymes. SPD has been shown to enhance cellular resilience and promote the expression of key enzymes involved in steroidogenesis, thereby supporting testosterone synthesis even in the presence of inflammatory stimulus. These findings highlight the intricate balance between inflammation and steroidogenesis, where compensatory mechanisms may be activated to preserve testosterone production. Further studies are needed to elucidate the precise molecular pathways involved in this regulation.
Comment:
- Lines 210-237: The transcriptomic analysis does not provide the specific names and functional annotations of key differentially expressed genes (DEGs). The conclusions are only summarized through pathway enrichment, and a list of key genes should be supplemented (can be included in the supplementary materials).
Response 9: We thank the reviewer for this suggestion. A list of key differentially expressed genes (DEGs) and their functional annotations has been added to the supplementary materials (Supplementary file 2). This provides a more detailed view of the transcriptomic data and supports the pathway enrichment analysis.
Comment:
- Lines 350-352: The feasibility of SPD as a feed additive is insufficiently discussed. References to studies on its application in livestock should be included to support this point.
Response 10: We appreciate the reviewer’s suggestion. In the revised manuscript, we have expanded the discussion on the feasibility of SPD as a feed additive, citing relevant studies on its application in livestock (citations added). This addition strengthens the practical implications of our findings.(Lines 486-509). Here’s the additions:
Our study further explored the feasibility of SPD as a feed additive and analyzed its potential applications based on existing research. Previous studies have suggested that exogenous SPD may promote growth, improve intestinal health, and enhance antioxidant capacity in livestock and poultry. For instance, SPD supplementation has been shown to significantly improve growth performance and reduce the incidence of diarrhea in pig-lets1. In geese, SPD was found to regulate the expression of polyamine metabolism-related genes in the small intestine, thereby maintaining intestinal homeostasis2. However, re-search on the application of SPD in ruminants remains limited, particularly regarding its effects on reproductive health under high-altitude conditions.
To further evaluate the safety of SPD, we first conducted a toxicity experiment using a mouse model in which we examined whether long-term administration of SPD-containing water (10 mg/kg for 1, 7, and 14 days) would lead to liver dysfunction, abnormal hematological parameters, or histological changes in organ tissues. The results showed that SPD treatment did not cause significant liver damage, and no abnormalities were observed in hematological parameters or histological examination of organ tissues. These findings suggest that SPD exhibits good biological safety within the tested dosage range, providing supporting evidence for its potential use as a feed additive.
However, the metabolic differences of SPD among different animal species require further investigation. Specifically, in high-altitude grazing yaks, the metabolic process, bioavailability, and long-term effects of SPD on reproductive health remain unclear. Fu-ture studies should focus on elucidating the specific mechanisms of SPD in ruminants and evaluating its feasibility in practical livestock production. This will provide a scien-tific basis for the potential application of SPD as a feed additive in economically important animals such as yaks.
Reviewer 3 Report
Comments and Suggestions for Authors
Manuscript entitled Spermidine as a Potential Protective Agents against Poly(I:C)-Induced Immune Response, Oxidative Stress, Apoptosis, and Testosterone Decrease in Yak Leydig Cells brings original data on an interesting research topic regarding the therapeutic potential of spermidine in protecting Leydig cells from oxidative stress, apoptosis, and functional damage induced by Poly(I:C). Namely, Abstract, Introduction, Results, Discussion and Conclusions were well defined. Article contains lots of interesting data and tables. The data were statistically processed well. References are adequate.
However, there is one minor flaw that should be addressed and revised before publishing.
Line 378 – 379. In The Materials and Methods section does not state how you obtained the testicles to perform the experiment.
Author Response
We sincerely appreciate the your insightful comments and constructive suggestions. Your comments and suggestions have been immensely helpful in improving the quality of our work. Below, we provide a response to the concern raised. For your convenience, the modification in the manuscript has been highlighted in yellow and the relevant changes are pasted under the response.
Comment:
Line 378 – 379. In The Materials and Methods section does not state how you obtained the testicles to perform the experiment.
Response:
Thank you very much for your thorough review and valuable feedback on our manuscript. Your comments and suggestions have been immensely helpful in improving the quality of our work. The testicles used in this study were obtained from three sexually mature male yaks (3–4 years old) raised at the Qingbaijiang Yak Farm in Chengdu, Sichuan Province, China. After slaughter, one testicle from each yak was transported to the Reproduction Laboratory at Sichuan Agricultural University for further processing and Leydig cell isolation.
To ensure transparency, we have added this information to the Materials and Methods section in the revised manuscript (Section 4.1, Lines 512–515). Here’s the addition:
The testicles used in this study were obtained from three sexually mature male yaks (3–4 years old) raised at the Qingbaijiang Yak Farm in Chengdu, Sichuan Province, China. After slaughter, one testicle from each yak was transported to the Reproduction Laboratory at Sichuan Agricultural University for further processing and Leydig cell isolation.
Round 2
Reviewer 1 Report
Comments and Suggestions for Authors
Authors have improved the manuscript based on my suggestions, I think it is suitable to accept
Comments on the Quality of English Languagecan be improved
Reviewer 2 Report
Comments and Suggestions for Authors
The authors have answered all my concerns.